# The Sometimes Context-Specific Habituation: Theoretical Challenges to Associative Accounts

**DOI:** 10.3390/ani11123365

**Published:** 2021-11-24

**Authors:** Yerco E. Uribe-Bahamonde, Orlando E. Jorquera, Edgar H. Vogel

**Affiliations:** 1Facultad de Psicología, Universidad de Talca, Talca 3460000, Chile; yuribe@utalca.cl; 2Centro de Formação em Ciências Ambientais, Universidade Federal do Sul da Bahia-UFSB, Porto Seguro 45810-000, BA, Brazil; ojorquerc@gmail.com

**Keywords:** habituation, context–specificity, contextual learning, priming theory, animal learning, SOP

## Abstract

**Simple Summary:**

When a stimulus occurs repeatedly without significant consequences, animals tend to decrease their responses to that stimulus. This phenomenon, known as habituation, can be explained by a class of theories that posit that expected events are less effective in provoking their responses than unexpected events. According to Allan Wagner’s priming theory, one of the ways this expectation might happen is via associative learning between the stimulus and the context in which stimulation occurred. In this article, we summarize a few theoretical complexities that derive from this approach along with some relevant empirical questions that remain open to further research.

**Abstract:**

A substantial corpus of experimental research indicates that in many species, long-term habituation appears to depend on context–stimulus associations. Some authors have recently emphasized that this type of outcome supports Wagner’s priming theory, which affirms that responding is diminished when the eliciting stimulus is predicted by the context where the animal encountered that stimulus in the past. Although we agree with both the empirical reality of the phenomenon as well as the principled adequacy of the theory, we think that the available evidence is more provocative than conclusive and that there are a few nontrivial empirical and theoretical issues that need to be worked out by researchers in the future. In this paper, we comment on these issues within the framework of a quantitative version of priming theory, the SOP model.

## 1. Introduction

When a stimulus occurs repeatedly, the predominant consequence is a systematic decrease in the frequency or amplitude of the response to that stimulus. If it is shown that this decrease is not caused by physiological changes at the sensory or motor level, it is inferred that a learning phenomenon, known as habituation, has occurred. Experimentally, habituation is demonstrated by repeating a single stimulus with no further consequence for the animal, thus it is almost unanimously classified as a non-associative form of learning [1,2].

In any experiment of habituation, however, animals are exposed to many other stimuli that are present in the experimental environment and that might become associated with the target stimulus. This possibility was entertained by Wagner [3,4,5] in his priming theory, which holds that the effectiveness of a stimulus in producing its response decreases when the stimulus is already signaled or predicted by other stimuli. Wagner supposed that this priming or prediction might come from the stimulation context which acquired a Pavlovian association with the target stimulus. Thus, when a stimulus is repeatedly presented in a context, the context would act as a conditioned stimulus (CS) that would develop an association with the habituating stimulus, which would play the role of an unconditioned stimulus (US). Under this premise, habituation is expected to be context specific, so that presenting the habituated stimulus in a context other than that of training should lead to a recovery of the response. Likewise, habituation should be extinguishable by exposing the animal to the habituation context in the absence of the habituated stimulus.

Dissegna, Turatto, and Chiandetti [6] reviewed this sort of experimental evidence and concluded that contextual control of habituation is a true phenomenon in a wide range of animals and response systems. They interpret the reviewed evidence as supportive of a priming theory and proposed to regard habituation as an associative form of learning (see also [7]). Although less emphasized by them, however, the literature also shows some negative evidence, which has been used by others to argue against Wagner’s view of habituation [8,9,10,11].

Because of this controversy, we think that there is reason to look more closely at the designs of experiments on contextual control of habituation. Secondly, we think that several quantitative subtleties of Wagner’s theory are underutilized in the field and a good part of this paper is devoted to promoting their consideration in future research. We propose that to fully appreciate the consequences of adopting Wagner´s theoretical approach, a precise definition of both associative and non-associative influences needs to be stated.

In this article, we summarize a few key theoretical and computational aspects of Wagner´s theory along with some relevant empirical questions that derive from them. We begin by summarizing the type of experimental procedures that have been used to examine contextual control of habituation based on the nomenclature offered by Dissegna et al. [6]. Then, we proceed by presenting the non-associative aspects of the theory, which are often occupied to describe the so-called short-term habituation, dishabituation, and sensitization phenomena [12]. Next, we show how the associative machinery of the theory can be adopted to explain long-term habituation [13]. We contrast these theoretical aspects with some of the evidence reviewed by Dissegna et al. [6] and outline the type of research that is suggested by these contrasts. We conclude that researchers still have some way to go before arriving at a definite conclusion with respect to the associative or non-associative nature of habituation.

## 2. Experimental Designs to Study Contextual Control of Habituation

In Pavlovian conditioning, to demonstrate that a CS–US association has been acquired, researchers must show that pairings of the CS and the US at time t1 are the cause of the changes in behavior at a subsequent time, t2. Although the ideal design for this demonstration is subject to some debate, many would recommend comparing a paired CS–US condition with a control condition, known as “explicitly unpaired control”, in which the two stimuli are presented for the same number of occasions, but separated in time [14]. Thus, to test the hypothesis that habituation is a product of context–stimulus associations, experiments must be designed to conform to this basic requirement. This is not an easy task, considering that contexts are very different from standard conditioned stimuli (in terms or duration, experimental control, distinctiveness, etc.) and that the focus of habituation studies is not on the development of a conditioned response, but on the conditioned diminution of the unconditioned response [15].

Consider, for instance, the explicitly unpaired control used in many studies of conditioning. In habituation, where the role of conditioned stimulus is played by a very long and poorly controlled set of contextual cues, the mere task of distinguishing between paired and unpaired conditions presents an almost insurmountable challenge. Indeed, strictly speaking, the unpaired condition is not viable in habituation, because it is impossible to present the habituating stimulus in the absence of any context at all. In the face of these restrictions, researchers have proposed other procedures, which, although less conclusive in isolation, might be very informative when taken together. Dissegna et al. [6] reviewed, summarized, and classified the evidence with each of these procedures, whose major features can be grasped from Figure 1. The figure sketches what it would be expected if context–stimulus associations were acquired. For the sake of simplicity, the figure presents an idealized panorama in which all conditions can be viewed as a continuous episode of time divided into four stages: Pre-habituation, habituation, post-habituation, and test. In the pre -and post-habituation stages, the animals are exposed to certain contexts with no presentation of the target stimulus. In the habituation stage, the target stimulus is presented repeatedly to ensure habituation (sketched in the figure as four trials, each with a decreased response relative to the previous trial). In the testing stage, retention of habituation is examined in just a few trials (cartooned in the figure as one trial).

The top plot of Figure 1 shows the paired condition, where the response decreases during habituation and remains somewhat diminished at test. This is the typical control condition with which each of the experimental manipulations is compared. The first is the so-called “context change” condition, where animals are exposed to the stimulus in a distinctive context A and then, after a delay, they are tested for responding to the same stimulus in a different context B. If the context shift results in a recovery of the habituated response in a way that does not in an unchanged-context condition, it is said that habituation is context-specific. Although recovery in responding after context shift is predicted by associative accounts of habituation, this manipulation does not rule out other possibilities, such as the potential sensitizing or distracting effects of the new context. To cope with these confounds, researchers have proposed a variation of the protocol, consisting of exposing the animals to the new context before testing. Here, the supposition is that a familiar context should produce a minimal of arousing and distracting effects when the stimulus is presented for the first time within it. These two possible designs denominated “context–change (novel) and context–change (familiar)” are illustrated in the second and third row of Figure 1.

In the Pavlovian conditioning literature, it is a well-established fact that the effects of CS–US pairings can be reduced if the CS is presented without the US before (latent inhibition) or after (extinction) conditioning. As seen in the last two plots of Figure 1, extinction and latent Inhibition procedures can be applied to habituation by exposing animals to the habituating context during the pre-habituation or post-habituation stages, respectively. The figure shows what would be expected in the test stage if the context behaved as a CS: More responding in the latent inhibition and extinction conditions relative to the control condition. These procedures can be conveniently combined with the context change procedure because none of them involves the presentation of a presumably arousing new context in test. As we shall demonstrate in the next section, however, the theoretical processes that might be involved in latent inhibition and extinction may differ from each other in important ways.

Of course, Figure 1 is an oversimplification for illustrative purposes. There are several ways in which these procedures can vary and thus, be differentially informative with regards to the hypothetical associative nature of habituation. For instance, out of a total of 60 studies reviewed by Dissegna at al. [6], the vast majority (47 studies) provided positive evidence of context-specific decrement in at least some of the procedures described in the figure. However, their review also showed some conflicting evidence, which, we think, might provide a clue about when contextual control of habituation should or should not be expected. Factors such as the type of response, amount of training, and frequency of stimulation seem to be critical (see [16], for speculations in this respect). The existence of these factors poses several challenges for the associative component of Wagner´s approach, but they also provide an opportunity to consider the non-associative aspects of the theory. In what follows, we elaborate on these considerations.

## 3. Priming Theory and the SOP Model

Priming Theory [3] states that the effectiveness of a stimulus to produce its response is diminished when there is a prior representation of such a stimulus in a limited-capacity active memory. This effect can be a consequence of a recent presentation of the stimulus (self-generated priming) or of retrieval of its representation from long-term memory by an associated stimulus (associatively generated priming). According to this view, the observed diminution in responding to a repeated stimulus or habituation would depend on a combination of a transient and non-associative effect, due to recent exposure to the stimulus, and on a persisting and associative effect due to the context acting as a conditioned stimulus. Wagner offered two further speculations: (a) An extraneous stimulus or distractor can produce dishabituation by displacing the representation of the target stimulus out of active memory, and (b) a stimulus that controls arousal effects can enhance the response to the target stimulus or sensitization. In principle, both dishabituation and sensitization are short-term and non-associative influences that can be exercised by several kinds of stimuli, including contextual cues.

Although these notions can in principle describe many observations resulting from stimulus repetition, it is important to state more precisely how these associative and non-associative processes work in isolation and in tandem with each other. A quantitative elaboration of priming theory, known as the sometimes-opponent processes model (hereafter “SOP model” [17,18]) may be useful for this purpose. Thus, we next proceed to describe the general principles of the SOP model and its potential in accounting for and guiding habituation studies. Our intention here is to provide readers with a flavor of how SOP operates, without going into quantitative details. For a more detailed exposition of the theory, the reader may consult Mazur and Wagner [18], Uribe-Bahamonde et al. [13], Uribe-Bahamonde et al. [19], Vogel, Ponce, and Wagner [20], and Wagner [17]. A full quantitative description of the model and an open access simulator can be found in http://vogelab.com/prohabituationlab (accessed on 20 October 2021).

Figure 2 presents a simplified rendition of the theoretical processes involved in a typical habituation experiment according to SOP. The model assumes that the processing of the habituating stimulus can be characterized in terms of the distribution of its theoretical elements across three states of activity: Inactive (I), primary (A1), and refractory (A2). It is supposed that when the stimulus is turned on, some of the inactive elements are moved to primary activity at a rate of p1, from which they subsequently decay, first to the refractory state, at a rate of pd1, and then back to inactivity, at a rate of pd2, where they remain unless a new presentation of the stimulus occurs. SOP assumes that the presentation of the stimulus provokes a first component of the response, which depends on the proportion of elements in the A1 state of activity, and then a second component, dependent on the proportion of elements in the A2 state. Depending on the response system, the second component can be agonist, antagonist, or unrelated to the first component of the response. For the sake of simplicity, and to show that several facts of habituation can be derived from these very simple principles, let us assume, for now, that the response of interest depends uniquely on A1 activity. Finally, SOP assumes that processing of the target stimulus is not only influenced by the stimulus itself, but also by contextual cues in three very different ways: Dishabituation, short-term sensitization, and associative priming. We will elaborate next on each of these theoretical aspects.

### 3.1. Non-Associative Refractory-Like Effects According to SOP

To illustrate how the model operates, let us ignore for the moment the effects of the context and focus on the theoretical processes provoked by the target stimulus alone. Figure 3a depicts the distribution of elements in the A1 and A2 states over time in a simulated session of habituation in which four stimuli were presented at a fixed short interval and when the stimulus was presented a fifth time in a test session occurring after a certain delay. 

Let us consider first what happens in the first four trials: The plot show that the presentation of the stimulus produces a rapid and transitory increase in the proportion of the elements in the A1 state, reaching a peak equal to the p1 value (p1 is constant throughout the duration of the stimulus and that is represented in the figure as a white rectangle). A1 activity is followed by an increase in the proportion of elements in the A2-sate and by a delayed return of elements to inactivity. It is evident that after the first presentation of the stimulus, and when p1 and A1 have decayed towards zero, there is still some time in which a substantial proportion of elements are in the A2-state, thus they could not be promoted to the A1-state when the stimulus is presented again. Thus, since the response depends on A1 activity, the second presentation of the stimulus is less effective in provoking the response that the first presentation. As seen in the figure, this effect accumulates over time and across trials up to the point that by the fourth trial, the A1 activity occasioned by the stimulus is decremented by the remnant A2-activity caused by all prior trials.

Notice that with this very simple mechanism, SOP predicts not only a progressive decrement over trials, but also that the habituated response recovers to some degree if time is allowed to pass without stimulation. This last phenomenon is called “spontaneous recovery” and refers to the observation of a partial recovery of a habituated response within minutes, hours, or days after the habituation episode [21]. In the model, this is represented by the fact that self-generated priming loses its effect as the time from the last trial transpires. This effect can be appreciated in Figure 2a, which shows that the amplitude of A1 in the fifth trial is somewhat recovered relative to the fourth trial but still lower than the first trial. Of course, this is a transient effect that would disappear if the habituation–test interval was increased. The fact that spontaneous recovery increases as the retention interval increases has been shown in a few habituation protocols such as the startle response of rats [22], the escape response of crabs [23,24], the gill withdrawal reflex of the aplysia [25], the escape response of C-elegans [26,27], and proboscis extension in Honeybees [28]. Other reports, however, have found null results [29,30].

Following this reasoning, then, one possibility that should not be discarded a priori is that the partial retention of habituation seen in some protocols could be due to a non-associative mechanism. The critical empirical question is: How long did this non-associative influence last? On the theoretical side, the SOP model predicts that the size and duration of self-generated priming depend on several quantitative considerations, such as the absolute value of p1, which can be assimilated to the intensity of the stimulus, and the relative size of the parameters pd1 and pd2, which dictates the duration of the A1 and A2 processes, respectively. Although we will not proceed further in this respect, it is important to emphasize the fact that the model has a non-associative mechanism that could describe some instances in which the recovery of the response is insensitive to changes in the context, because the observed decrement does not depend on context–stimulus associations. This fact can be particularly suggestive in studies showing no context specific habituation with short retention intervals [10,31,32,33,34,35].

### 3.2. Two Potential Non-Associative Effects of the Context: Dishabituation and Short-Term Sensitization

A very important, but often ignored assumption of the SOP model is that there is a limit in the total amount of A1 and A2 activities across all stimulus representations. This is represented in the model as an increase in the decay rates (pd1 and pd2) proportional to the aggregate A1 and A2 activity across all stimuli at a given moment. Thus, if while the target stimulus is being processed, an extraneous stimulus or distractor is presented, SOP assumes that the decay rates pd1 and pd2 of the target stimulus will be increased with the consequent acceleration of the return of its elements to inactivity. Thus, thanks to the distractor, the elements of the target stimulus will be released from self-generated priming and will be available for activation sooner. This competition for a limited processing capacity would be responsible for dishabituation according to SOP.

Dishabituation is commonly demonstrated by interposing an innocuous stimulus between two presentations of the target stimulus resulting in a transient recovery of the response [7,36]. Although it is predominantly demonstrated by using an explicit stimulus, it is conceivable that tonic cues, like contexts, might have a similar effect. Furthermore, it is altogether reasonable to suppose that novel contexts are more effective distractors than familiar contexts. This is represented in Figure 2 as a pair of dotted arrows connecting the novel context B to the pd1 and pd2 parameters of the target stimulus. The implication of this idea for habituation is sketched in Figure 3, which shows that SOP predicts a greater recovery in the response in testing in the context change condition (panel b) relative to the paired condition (panel a). This outcome results from the application of the distractor rules described above (note that in panel b, A2 is completely obliterated by the presentation of the new context in test). It is conceivable, thus, to describe some instances of context-specific habituation without appealing to an associative process.

Of course, the dishabituating effects of a novel context are restricted to those cases in which self-generated priming is presumably still in operation at the time of testing. This might well be the case of the studies in which the habituation–test interval was very short [10,31,32,33,34,35,37].

There is, however, a further way in which the response to the target stimulus can be incremented by non-associative influences: Sensitization. This term has been used since long to explain the fact that, if certain conditions are met, the behavioral decrement that normally follows stimulus repetition might be delayed, reduced, restored, or even replaced by a transient increment in the response [38]. In SOP, the possible sensitizing effect of a novel context can be implemented by assuming that presentation of a given context—especially one that has arousing properties—enhances the activation parameter, p1, of the target stimulus (see Figure 2). By this assumption, the response to the target stimulus is expected to be potentiated by the context. Of course, again, it is reasonable to assume that novel contexts have more arousing properties than familiar contexts, which is illustrated in panel (c) of Figure 3. It can be seen that a novel context might not only provoke dishabituation of the type already described in panel (b), but it might also provoke an increase in p1, and hence an even greater increase in the response.

In sum, because of these potential non-associative influences of novel contexts on the response to the target stimulus, it is very important to consider the “context change (familiar)” condition described in Figure 1. Unfortunately, many of the studies in which context change have produced a “recovery” of the habituated response, have not used this condition, leaving, therefore, some interpretative ambiguity (see [1]).

### 3.3. Associatively Generated Priming According to SOP

As shown in Figure 3, the non-associative mechanisms of self-generated priming can explain retention of habituation in those circumstances in which the habituation–test interval is relatively short. Although the meaning of “short” should be empirically determined and surely vary across different species depending on their respective life cycle, it seems implausible that all instances of retention of habituation can be explained through this process. Indeed, most researchers agree that habituation cannot be reduced to a refractory-like effect [39]. Furthermore, when using the expression “non associative form of learning”, most researchers focus on what is called “short-term habituation”, which roughly refers to the decrements in the response that occur within a session of stimulation, part of which spontaneously vanishes with the passage of time [38]. Perhaps the more convincing evidence in favor of this distinction comes from findings that intrasession and inter-session decrements are uncorrelated [40] and that they appear to depend on different neurobiological mechanisms [41].

In line with this distinction, SOP assumes that long-term habituation depends on the development of an association between the context and the target stimulus. As shown in Figure 2, context A, via its associative link, influences the processing of the habituating stimulus by promoting elements directly from the inactive state to the secondary state at a rate of p2. By this action, the context has an indirect behavioral consequence, which is decreasing the Al activity occasioned by a subsequent presentation of the target stimulus, and thus decreasing its primary response. If it is assumed that this association develops on a trial-by-trial basis at a relatively slow rate and that its expression requires a consolidation time, it is conceivable that its predominant effect would be seen in pure form in a delayed retention test. This is illustrated in Panels (a) and (b) of Figure 4, which depict the theoretical processes that would be involved in the paired and context change conditions when a very long habituation to test interval is assumed. As can be seen in panel (a) some elements of the target stimulus are promoted to the secondary activity state by the precedence of the habituating context A, so that there is less primary activity when the stimulus is presented again in test. On the contrary, no effects are expected in the context change condition, in which the response at test is identical to that of the first trial of the habituation session. Since in this case context B is a familiar context, no sensitization nor dishabituation are assumed in this condition. The increment in the response would be the result of the mere passage of time and of the change in context in the context–change condition.

At an intuitive level, context–stimulus associations seem to provide a reasonable explanation of LTH which is also consistent with the few studies that have demonstrated context specificity using a familiar context B. However, it might be convenient to take a closer look at how this association is formed according to the model. Virtually any theory of associative learning posits that temporal contiguity between CS and US is critical for the acquisition of an association between them. Such theories also assume that an acquired association is extinguished if such a contiguity is subsequently broken, for instance by presenting the CS without the US. SOP brings the idea of contiguity to a greater extent by computing acquisition and extinction on a moment-by-moment basis. Thus, the relative distribution of elements in the three states of activity over time and across the experimental cues is critical for what is learned in an episode of conditioning. In the case of habituation, it would be necessary to assume that the context is represented by its own set of elements that can be in one of three activity states. Since the context is turned on at the beginning of the habituation session and off at its termination, it can be assumed that there is a relatively constant number of elements in the A1 state throughout the session [42]. Once the respective representations of the context and the target stimulus have been established, associations between them can be computed.

According to SOP, context–stimulus associations are assumed to be the result of excitatory minus inhibitory associations that develop simultaneously depending on the respective states of activity of the context and the stimulus. The acquisition of an excitatory association is the product of concurrent A1 activity of the context and the stimulus. In contrast, the acquisition of an inhibitory context–stimulus association is the product of concurrent A1 activity of the context and A2 activity of the stimulus. The net association is computed by subtracting inhibitory associations from excitatory associations. If the net association is excitatory, it will endow initial activity of context to provoke the movement of elements of target stimulus from inactivity to secondary activity at a rate of p2, which is proportional to the net association. If p2 is negative or zero, of course, there is no direct influence of the context on the representational activity of the target stimulus. In simple terms, p2 will increase as a consequence of the context overlapping with A1 activity of the target stimulus and will decrease as a consequence of the context overlapping with A2 activity of the stimulus. It follows then that if a context is presented alone during the post habituation session, its own ability to produce A2 activity in the absence of A1 activity will provoke a progressive predominance of inhibitory learning and hence extinction of the association. This effect is illustrated in panel (c) of Figure 4. A good number of studies have demonstrated this effect in habituation protocols [16,23,27,43,44,45,46,47,48].

It should be recognized, however, that some authors have pointed out some fundamental problems of conceiving contexts as conditioned stimuli. For instance, Vogel et al. [42] demonstrated that contexts cannot be represented merely as long-duration conditioned stimuli, because no net contextual learning would occur due to the context being exposed to long periods of extinction during the inter-trial intervals. This is a general problem for associative theories that, like the SOP model, compute acquisition during trials and extinction during the intertrial intervals. Vogel et al. [42] proposed that this dilemma can be solved by assuming that the context is represented by a series of components that form separate associations with the target stimulus. They propose that the activation of these elements is governed by a random process over the entire session, except when the target stimulus is presented, an occasion in which the activity of the components is progressively suppressed. This fact biases learning toward producing more excitatory learning. An examination of the potential of this approach is beyond the scope of the present paper, thus interested readers may consult Vogel et al. [20,42]. 

A second theoretical problem of SOP´ s conception of LTH is the so-called latent inhibition phenomenon. In Pavlovian conditioning, the term “latent inhibition” is used to describe the observation that contextual stimuli that had embraced CS presentations, subsequently diminish the capability of the CS to get associated with the US when the CS–US pairings occur in that same context. SOP explains this phenomenon readily by assuming that the context and the CS become associated during the first phase, and, as a consequence of this, the CS gets primed by the context in the second phase. Priming causes a decreased A1 processing of the CS (or LTH) leading to lesser excitatory learning when the CS is paired with the US in the second phase. This explanation, however, is implausible when the context itself occupies the role of the CS, as is the case of the design depicted in Figure 1e. SOP does not anticipate a diminished representation of the context by merely exposing the animals to it prior to habituation. As shown by Dissegna el al. [1], the recovery of a habituated response after a latent inhibition manipulation has been reported in several studies [23,27,43,44,45], but there is also negative evidence [49]. Should this effect probe be replicable, further theoretical assumptions must be added to SOP. Another possibility is to consider entirely different approaches such as that of Hall and Rodriguez [9].

### 3.4. Multiple Associative Influences of the Context

In the previous sections, we showed that when testing the level of response to a previously habituated stimulus, SOP predicts that the context in which this test occurs might have several associative and nonassociative influences. The nonassociative influences, namely dishabituation and short-term sensitization, are incremental and are more likely to be exercised by unfamiliar contexts. On the contrary, the associative influence, namely associative priming, is decremental and is exercised by the specific context in which habituation occurred. Thus, when comparing responding in same versus different context conditions, it is essential that animals have some degree of familiarity with the shifted context to minimize the confounding effects of the non-associative influences of novel contexts.

Wagner and Vogel [50] pointed out, however, that the theoretical principles of SOP and the empirical evidence seem to call for an even more complex approach. They proposed that the associative influence of the context might include response-incrementing tendencies in addition to the decremental effects. These incremental effects can take two forms: One is what they called “CR-contribution to the measured response’’ referring to the fact that the conditioned response evoked by the context might add to the unconditioned response of interest evoked by the target stimulus. The second tendency, called “emotive potentiation of the response” refers to a more generalized conditioned emotional response that resulted from pairing the context with some arousing aspects of the target stimulus. We will refer to the latter effect as “long-term sensitization”. Figure 5 summarizes each of these tendencies that we shall comment in further detail next.

To understand the theoretical possibilities of the CR-contribution effect, let us return to the issue that we left open regarding response generation in SOP. As shown in Figure 5, the core supposition is that the presentation of the target stimulus provokes a two-component sequence of responding, first being dependent on A1 activity and subsequently on added A2 activity. As discussed in previous sections, the habituating context (A), via its associative link, has an indirect influence on behavior by provoking A2 activity that decreases A1 activity provoked by a subsequent presentation of the target stimulus (i.e., LTH). Note however, that by this same action of causing A2 activity, the context also has a direct behavioral consequence, which would depend on whether the behavior occasioned by A2 activity is similar, opposite, or unrelated to the behavioral consequence of A1. One way of simplifying this panorama is by defining a composite response rule. Wagner [17] proposed that the response to the target stimulus is a function of the sum of the number of elements in the A1 and A2 states of activity weighted by the linear constants w1 and w2, respectively. This assumption brings forth three very different scenarios regarding habituation [51].

The first scenario is assuming that w2 equals zero, as we have done so far in this paper. In this case, the response would depend entirely on A1, with A2 contributing only indirectly via its priming effect on A1. In the examples of Figure 3 and Figure 4, we have adopted this tactic, because it is consistent with the majority of the response systems in which contextual control of habituation has been studied (e.g., orienting, startle, suppression, and scape; see [1]). In this case, then, the associative role of context A would be exclusively decremental. The second alternative is to adopt a negative value for w2, in which case the secondary activity is subtracted from the primary activity to produce the response. Here, both the negative contribution of A2 to the response and its priming on A1 would act in a synergic way to diminish the primary response to the stimulus. The use of w1 and w2 with opposite signs may be particularly advised when there are empirical reasons to believe that the response to the habituating stimulus shows a secondary response that opposes to the primary response, as it has been frequently reported with pharmacological stimuli [52]. The third theoretical alternative is to assume that w2 is positive. Here, A2 would have two opposite effects on the response: An augmentative effect through summation with A1 and a diminutive effect through priming. In this more complex scenario, it would be expected to observe less behavioral habituation than in the former cases.

Things can get even more complicated if we consider that the majority of the stimuli evoke more than just a single response. A particularly interesting case is that any theoretical and empirical analysis of habituation as an associative process should regard the additional influences of emotional responses that can also be conditioned to contextual cues, and thus to potentiate the response to the habituating stimulus. This is what Wager and Vogel [50] referred to as “emotive potentiation of the response”. For instance, a habituation procedure involving an aversive stimulus such as, for instance, a loud noise, would lead the contextual stimuli to provoke associative priming (LTH) to that specific stimulus, but also a conditioned emotional response that would potentiate the response to any other stimulus, including the one that was habituated. Theoretically, there are two added assumptions of SOP that allow for such response-potentiating effects as well as decremental–habituation effects (see [13,53]). The first is that the presentation of the target stimulus activates two separate sets of A1/A2 units, one representing the sensory–motor aspect of stimulus and the other its emotional–arousing aspect. The association of the context with the sensory–motor aspect of the stimulus is assumed to influence the stimulus-elicited response, whereas the association of the context with the emotive aspect of the stimulus is assumed to control a generalized conditioned emotional response. The second assumption is that the emotive A2 activity modulates the parameter, p1, of stimuli experienced in its presence. In the current example, the processing of any stimuli would be potentiated by the conditioned emotional response developed by the context associated with the target stimulus.

Thus, the combination of sensory–motor and emotive associations of the context with the target stimulus led to stimulus-specific decrements and generalized increments in the target behavior. Moreover, the target response might also be modulated by the incremental, decremental, or null contribution of the conditioned response caused by the context (via w2). Although these possibilities reveal much of the complexity of the SOP model, they also show its tremendous explanatory potential. Remember that the emergence of the A2 process is at the heart of priming, and that the associative effect of the context is precisely to evoke A2 activity in the processing of the target stimulus.

Following this reasoning, it should be clear that studies of contextual control of habituation must be designed so as to disentangle the response-potentiating (i.e., long-term sensitization) from the response-diminishing (long-term habituation) associative effects of stimulus repetition, both of which can be context specific. If long-term habituation is viewed as being relatively specific to the exposed stimulus, and long-term sensitization as being more globally influential, then even if they are both associatively mediated and context dependent, one should be able to separate the influences by tests, not only of the exposed stimulus, but of other potentially effective stimuli. Unfortunately, none of the experiments reviewed by Dissegna et al. have been designed in ways that would allow one to separate such effects.

## 4. Conclusions

This article is an elaboration on Dissegna et al. [1]’s review concerning research on context specific habituation, in which they concluded that habituation should be taken as an associative form of learning and that [3,4,5]’s priming theory is the best suitable theoretical approach. In this paper, we use a quantitative version of priming theory, the SOP model, to expand on these conclusions and to show that in studies of contextual control of habituation the original and shifted contexts might exert several associative and non nonassociative influences in the target behavior.

Dissegna et al. [1] conveniently grouped positive and negative evidence of contextual control of habituation according to the experimental strategies involved. We used this grouping to make several theoretical points that should lead to caution in interpretation of the studies. We sustain that some results that are commonly taken as indicative of an associative process can be also understood as non-associative influences, such as dishabituation and short-term sensitization which are provoked by the shifted context in the so-called “context change procedure”. We did this not so much with the intention of denying that context–stimulus associations may even occur, but to emphasize that the designs need to be sensitive to distinguish associative from non-associative influences. The use of what we called “familiar context change condition” is one way of reducing although not eliminating the confounding nonassociative effects.

If, in addition to showing recovery of a habituated response when the stimulus is presented in an equally familiar context, it is shown that the response recovers when the habituating context is extinguished in the absence of the stimulus, the context–stimulus association hypothesis might be greatly strengthened. This sort of double demonstration of context specific acquisition and extinction of habituation has been reported in the escape response of crabs [23,43], the orienting response of rats [16], and the escape response of nematodes [27] and worms [49]. Note that SOP does not predict a recovery in the response with the latent inhibition procedure (i.e., after pre-exposition to the context), which agrees with some negative evidence in earthworms [49], but not with positive evidence in crabs [23,43,44] and nematodes [27,45].

A further reason for concern when interpreting research on contextual control of habituation is the potential influence of emotional responses that can also be controlled by the habituating context. There are a plethora of studies showing that responding to certain stimuli can be enhanced when animals are placed in a context that has been independently paired with aversive stimuli. Although there is general agreement that decremental and incremental processes coexist in protocols of stimulus repetition, the potential context specificity of both is a unique prediction of Wagner’s theory. Of course, this general potentiation effect can complicate the interpretation of those studies that have failed to show context-specific habituation of certain responses [10,16,31]. In this case, having a procedure with demonstrated stimulus specificity seems to be critical.

Table 1 presents a summary of the behavioral effects that might be attributed to the context according to the SOP model. Each row of the table represents each of the five theoretical mechanisms discussed in this paper, namely dishabituation, short-term sensitization, associative priming, CR-contribution, and long-term sensitization. Each column represents one the five types of contexts in which responding to the target stimulus has been assessed in studies of contextual control of habituation (see Figure 1). To separate stimulus specific versus generalized effects, there are separate rows for a test with the target stimulus and a test with a novel or non-habituated stimulus. Each cell of the table is filled with one of four possible behavioral effects of the context, defined by whether response to the respective stimulus is expected to be reduced, enhanced, or restored, relative to its initial level. The table supposes that the test is conducted after the habituation session and at a time when the effect of self-generated priming is no longer influential, so that pure contextual effects are operative.

We shall not repeat here what we discussed in the main body of the paper. It just suffices to say that Table 1 can be used as an approximate theoretical heuristic to guide interpretation and design of studies of habituation. For instance, designs testing associative effects of stimulus repetition must consider the routine use of equally familiar testing contexts and multiple test stimuli. In this respect, procedures in which robust stimulus-specific habituation has been demonstrated, such as the escape response of crabs, rats and nematodes should be preferred over others, such as the startle response of rats and humans in which there is not clear evidence of such a specificity. Moreover, notice that according to SOP, the effects of context–stimulus associations can be overcome by post habituation exposures to the context (extinction), but it cannot be precluded by context pre-exposition (latent inhibition). This is a critical prediction that differentiates SOP from other accounts [9] that surely require further investigation.

One finding that particularly challenges the generality of the associative explanation of long-term habituation is the observation that the habituation of different responses to the same stimulus can be differentially sensitive to context change. In two experiments, Jordan et al. [16] simultaneously measured lick suppression and startle in rats and found that although both responses habituated, only lick suppression showed a significant recovery with context shift. Pinto et al. [31] reported similar effects by showing context-specific habituation of heart-rate acceleration, but not of the startle response in humans. These findings are consistent with a series of experiments by Marlin and Miller [10] who found no evidence of contextual control of habituation of the startle response. To explain these discrepancies, Jordan et al. suggested that perhaps the time window in which the startle response develops (latency of initiation and peak of 10 and 50 milliseconds, respectively) might be too short for the development context–stimulus associations. It is noteworthy that robust stimulus-specific habituation has been found with other responses, such as orienting [16], suppression [16], escape [23,33,43], vasoconstriction [3], visual capture [46], and heart acceleration [31], which are measured over windows of the order of hundreds of milliseconds to seconds. In principle, this variability might be captured in the SOP model by assuming that the longer the response, the greater the opportunities for conjoint processing of the representations of the context and the stimulus. This assumption would imply that there are several pairs of A1/A2 nodes, each representing different responses to the same stimulus and each possessing its own temporal parameters (e.g., pd1 and pd2). Although promising, the possible role of response duration on context–stimulus associations has not been further examined neither theoretically nor experimentally.

The most interesting aspect about the priming theory (and the SOP model) is that it provides an articulate way of thinking about the multiple influences that might be operating when animals change their response after repeated stimulation. The question is not whether habituation is or is not an associative form of learning, but how associative and non-associative influences interact. As we discussed in this article and summarized in Table 1, despite the long empirical tradition of the study of habituation, there are still many questions that are still waiting for empirical data.

## Figures and Tables

**Figure 1 animals-11-03365-f001:**
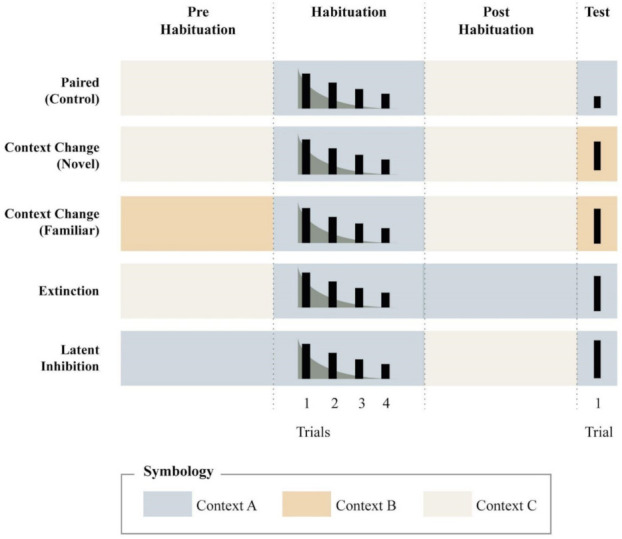
A simplified characterization of the main experimental designs to study contextual control of habituation. It is assumed that experiments comprise four phases: prehabituation, habituation, post-habituation, and test. In the pre and post habituation phases, animals are exposed to a given context without presentation of the target stimulus, while in the habituation and test phases, the stimulus is presented. The figure depicts five different groups or conditions which differ in the context in which some stages occur. The size of the black rectangles represents the amplitude of the response in each trial.

**Figure 2 animals-11-03365-f002:**
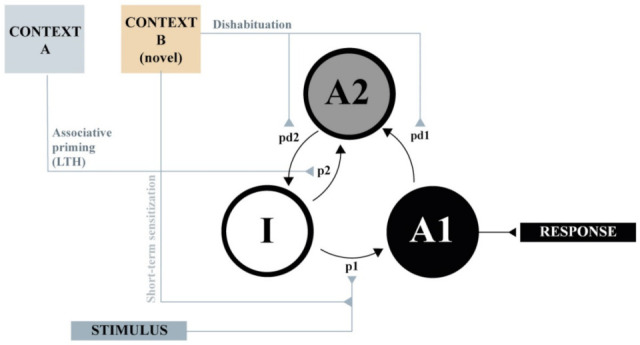
A network representation of the SOP model. The target stimulus is represented by a set of elements that can be in one of three states of activity: Inactive (**I**), primary (**A1**), and secondary (**A2**). The presentation of the stimulus leads to the promotion of elements to the A1 state according to the probability p1, from which they decay first to A2, with probability pd1, and then to I, with probability pd2. The habituating context A influences the activity of the target stimulus unit by means of the associative link p2 (long-term habituation). The novel context B influences the activity of the target stimulus by increasing pd1 and pd2 (dishabituation) or by increasing p1 (short-term sensitization).

**Figure 3 animals-11-03365-f003:**
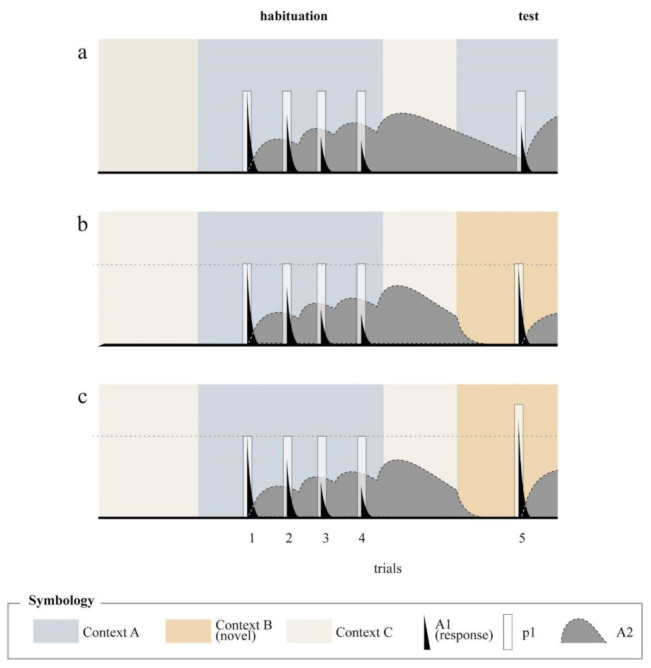
Theoretical processes at different stages of habituation procedures according to the SOP model. Panel (**a**) presents the predicted outcome for the paired condition and panels b and c, the outcome of the context–change (novel) condition. Panel (**b**) represents the prediction when a dishabituation effect of the novel context is assumed; and panel (**c**), when both dishabituation and sensitization effects are assumed.

**Figure 4 animals-11-03365-f004:**
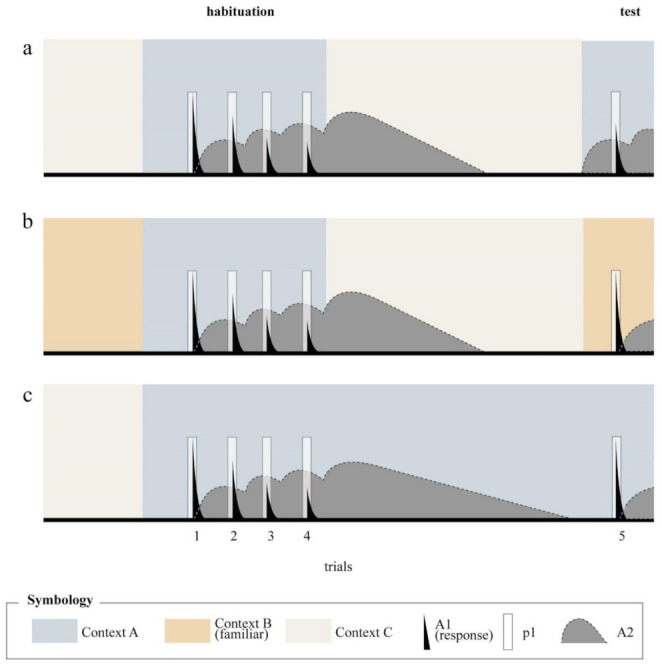
Theoretical processes at different stages of habituation procedures according to the SOP model with associative priming incorporated. Panel (**a**) presents the predicted outcome for the paired condition, panel (**b**) for the context–change (familiar) condition, and panel (**c**) for the extinction condition. Notice that in Figure 4, the habituation to test interval is considerably longer than in Figure 3.

**Figure 5 animals-11-03365-f005:**
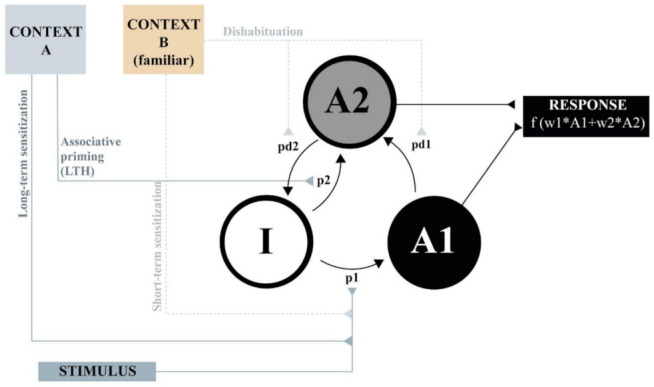
A network representation of the SOP model incorporating long-term sensitization and the response rule. The figure summarizes Wagner and Vogel (2010)’s view that the habituating context A might control three associative effects, one decremental, namely associative priming (LTH), and two incremental, namely conditioned response contribution to the measured response to the target stimulus, and long-term sensitization of the response to the target stimulus. The faint lines connecting context B with the representation of stimulus mean that short-term sensitization and dishabituation effects of this context can be discarded if a familiar context is used in the test.

**Table 1 animals-11-03365-t001:** Possible effects of the context in the retention of habituation test according to the SOP model.

Context	Stimulus	Non Associative Effects		Associative Effects				
		Dishabituation	Short-Term Sensitization	Associative Priming	CR Contribution			Long-Term Sensitization
					Agonist	Antagonist	Null	
Context A	Target	No effect	No effect	Decremental	Incremental	Decremental	No effect	Incremental
	Novel	No effect	No effect	No effect	No effect	No effect	No effect	Incremental
Context B (novel)	Target	Restoring	Incremental	No effect	No effect	No effect	No effect	No effect
	Novel	No effect	Incremental	No effect	No effect	No effect	No effect	No effect
Context B (familiar)	Target	No effect	No effect	No effect	No effect	No effect	No effect	No effect
	Novel	No effect	No effect	No effect	No effect	No effect	No effect	No effect
Context A (extinguished)	Target	No effect	No effect	Restoring	Restoring	Restoring	No effect	Restoring
	Novel	No effect	No effect	No effect	No effect	No effect	No effect	No effect
Context A (latent inhibited)	Target	No effect	No effect	Decremental	Incremental	Decremental	No effect	Incremental
	Novel	No effect	No effect	No effect	No effect	No effect	No effect	Incremental

## Data Availability

Not applicable.

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
