# Peer review of "The Sometimes Context-Specific Habituation: Theoretical Challenges to Associative Accounts"

_animals, 2021, doi:10.3390/ani11123365_

Round 1

Reviewer 1 Report

In general I think this is an excellent comment on the associative nature of habituation, which will nicely complement the review made by Dissegna et al. (2021).

I enjoyed reading the ms and I have no major concerns to raise. There is only one point that I think deserves the authors' attention, and that needs to be discussed in a revision. On page 15 the authors suggest that a factor that might determine whether or not habituation is context specific is the scale in which the specific response considered develops. They specifically conclude that responses measured on short time window might be insensitive to context change, whereas responses developing on longer time scale might lead to a context-specific habituation. I seriously doubt that this might be the case as, for example, Turatto and colleagues have repeatedly reported evidence of context-specific habituation for the attentional orienting response, which is extremely fast, in the order of a few hundreds of milliseconds. I invite the authors to discuss this issue.

Author Response

We thank the reviewer for this comment.  We acknowledge that we were too speculative in this paragraph in p. 15, so we rewrote it as follows (see lines 595-619):

“One finding that particularly challenges the generality of the associative explanation of long-term habituation is the observation that the habituation of different responses to the same stimulus can be differentially sensitive to context change. In two experiments, Jordan et al. (2000) simultaneously measured lick suppression and startle in rats and found that although both responses habituated, only lick suppression showed a significant recovery with context shift. Pinto et. al (2014) reported similar effects by showing context-specific habituation of heart-rate acceleration but not of the startle response in humans. These findings are consistent with a series of experiments by Marlin and Miller (1980) who found no evidence of contextual control of habituation of the startle response. To explain these discrepancies, Jordan et al. suggested that perhaps the time window in which the startle response develops (latency of initiation and peak of 10 and 50 milliseconds, respectively) might be too short for the development context -stimulus associations. It is noteworthy that robust stimulus-specific habituation has been found with other responses, such as orienting (Jordan et al, 2000), suppression (Jordan et al. 2000), escape (Hermitte, et al., 1999; Reyes-Jimenez, 2020; Tomisc et al, 1993), vasoconstriction (Wagner, 1976), visual capture (Turatto et al., 2019) and heart acceleration (Pinto et al, 2014), which are measured over windows of the order of hundreds of milliseconds to seconds. In principle, this variability might be captured in the SOP model by assuming that the longer the response the greater the opportunities for conjoint processing of the representations of the context and the stimulus. This assumption would imply that there are several pairs of A1/A2 nodes, each representing different responses to the same stimulus and each possessing its own temporal parameters (e.g., pd1 and pd2). Although promising, the possible role of response duration on context-stimulus associations has not been further examined neither theoretically nor experimentally.”

Reviewer 2 Report

Dissegna et al. recently published a review in this journal of empirical studies of context effects in habituation. In this they acknowledged that Wagner’s “SOP” theory could explain some, even many, of the effects described. On beginning to read the present paper I had hopes that it would explain in detail the way in which Wagner’s theory could deal with the various experimental results described by Dissegna et al.

In this I was disappointed. What Uribe-Bahamonde et al. offer is a very full and detailed account of the various complexities of the SOP theory as applied to contextual factors in habituation. There is nothing novel in this – all can be found in Wagner’s own writings – but they do a good, and useful, job of spelling out these complexities. What they do not do is to relate the various predictions that can be derived from SOP to the empirical results described by Dissegna et al. The paper would be rendered much more useful if they were to do so.

But perhaps we should simply be thankful for what we get. This paper does, after all, provide a succinct summary of how SOP applies to habituation and, in so far as I can tell, an accurate one.

On a number of occasions the authors depart from idiomatic English – but presumably the journal has editorial staff who will help with that.                        

Author Response

The reviewer is correct with respect to the limited scope of the paper. To further clarify this, we added the following paragraph in p 4-5, lines 179-184:

“Thus, we next proceed to describe the general principles of the SOP model and its potential in accounting for and guiding habituation studies. Our intention here is to provide readers with a flavor of how SOP operates, without going into quantitative details. For a more detailed exposition of the theory, the reader may consult Mazur and Wagner (1982), Uribe-Bahamonde et al. (2019), Uribe-Bahamonde et al. (2021), Vogel, Ponce, and Wagner (2019), and Wagner (1981)”.